# Effects of Modified Nano-SiO_2_ Particles on Properties of High-Performance Cement-Based Composites

**DOI:** 10.3390/ma13030646

**Published:** 2020-02-01

**Authors:** Zhidan Rong, Mingyu Zhao, Yali Wang

**Affiliations:** 1School of Materials Science and Engineering, Southeast University, Nanjing 21189, China; zmy8343@163.com (M.Z.); 220181887@seu.edu.cn (Y.W.); 2JiangSu Key Laboratory of Construction Materials, Nanjing 211189, China

**Keywords:** modified nano-SiO_2_ particles, high-performance cement-based composites, hydration, dispersibility, microstructure

## Abstract

In this research, silane coupling agent was used to modify the surface of nano-SiO_2_, particles and the effects of modified nano-SiO_2_ particles on the mechanical properties of high-performance cement-based composites and its mechanism were systematically studied. The results indicated that the optimum modification parameters were a coupling agent content of 10%, reaction temperature of 65 °C, and reaction time of 8 h. Compared with the unmodified nano-SiO_2_, the modified nano-SiO_2_ promoted and accelerated the hydration process of cement. The pozzolanic effect, filling effect, and nucleation effect of modified nano-SiO_2_ made the microstructure of the composite more compact, and thus improved static mechanical properties of cement-based composites.

## 1. Introduction

With the rapid development of global economy and the continuous advancement of large-scale infrastructure projects, more and more requirements are put forward for the strength, toughness, safety, and durability of concrete building materials. High-performance concrete (HPC) materials have been widely used in high-rise, long-span, ocean, protection, and harsh environment projects due to their excellent mechanical properties, durability, and safety reliability [1,2,3,4,5]. With the increasing demand for high-performance concrete, it is necessary to reduce porosity, optimize pore structure and improve compactness in the control of concrete performance. Therefore, adding nano-materials into cement-based materials is another way to improve the micro-structure of cement paste and pore structure from the micro-perspective in order to improve the mechanical properties of HPC [6,7,8,9,10]. 

Nano-materials have attracted much interest in cement-based materials during the past decade. At present, nano-SiO_2_ particles are the most widely used nano-materials in concrete. Numerous researchers have investigated the effects of nano-SiO_2_ particles on the performance of cement-based materials. According to previous research [11,12,13,14,15], due to the surface effect of nano-SiO_2_, nano-SiO_2_ particles could provide a lot of active specific surface area, which can be used as nucleation sites of pozzolanic reaction of cement hydration or active mineral admixtures to accelerate the early reaction. The small size of nano-SiO_2_ particles can fill the pore size between 10–100 μm, increase the bulk density of matrix, and improve the pore structure of cementitious materials. Rong [16] investigated the effect of nano-SiO_2_ on the hydration process and microstructural constituents of ultra-high-performance cementitious composites. The results shown that the hydration process was accelerated by the addition of nano-SiO_2_ and the microstructure was more homogenous and dense for nano-SiO_2_ specimens as compared to the control specimen. Liu [17] studied the effects of nano-SiO_2_ on early strength and microstructure of steam-cured high volume fly ash cement system. The results indicated that nano-SiO_2_ could largely increase the early strength of the high volume fly ash mortar. Fillability of nano-SiO_2_ and more C–S–H gel resulting from pozzolanic reaction of nano-SiO_2_ also decrease the porosity and refine the pore structure. Collodetti [18] investigated the potential of siloxane surface modified nano-SiO_2_ (nS) as an admixture capable of modifying heat development and microstructure densification of Portland cement pastes. The results shown that siloxane cover the nS surface and that the treatment allowed nS particles to last longer on a calcium hydroxide saturated solution before forming C–S–H; however, the particle size distribution of nS was not significantly affected by the procedure. 

Nano-materials can modify the microstructures and improve the mechanical properties of ordinary concrete or HPC to a certain extent. However, nano-materials have high surface energy and their particles are easy to agglomerate, so their reinforcement effect is difficult to give full play to. In this paper, a simple and effective surface modification method was used to modify nano-SiO_2_ to improve the dispersion of nano-SiO_2_, especially suitable for large scale production. The effects of modified nano-SiO_2_ particles on the properties of high-performance cement-based composites and their mechanism were also studied.

## 2. Experiment 

### 2.1. Materials 

HPC blended with cement, fly ash (FA), and nano-SiO_2_ (NS) were prepared in this research. The cement used was P·II52.5 in accordance with the relevant Chinese standard. The properties of P·II52.5 cement are similar to CEM I 52.5 according to European Standard <EN 197-1:2011>. The chemical compositions of both the cement and FA are shown in Table 1. The average particle size of NS was 20 nm and the purity was no less than 99.5%. Besides, the continuous grading river sand with a maximum particle size of 2.36 mm was used for mortar mixture, and the specific gravity and the fineness modulus of the sand were 2.6 and 2.26, respectively. One kind of polycarboxylic-type high-performance water reducer was adopted as the superplasticizer and the water reducing ratio was no less than 35%. 

### 2.2. Methods

#### 2.2.1. Surface Modification of Nano-SiO_2_

Firstly, nano-SiO_2_ suspension with ethanol as solvent was prepared. Then, the prepared suspension was ultrasonic cleaner for 15 min. In order to avoid the heat evaporation of ethanol, fresh-keeping film was applied to cover the bottle mouth. The PH value of the solution was adjusted to 4 by hydrochloric acid after ultrasound. After that, a certain amount of silane coupling agent (KH550) was added to the solution and stirred in a constant temperature magnetic water bath heater. The amount of silane coupling agent, reaction temperature, and reaction time were shown in Table 2. After the reaction, the suspension was filtered several times by a filter and washed with alcohol. Finally, the solid material was dried in a vacuum constant temperature drying chamber at 50 °C for 3 days. Thus, the modified nano-SiO_2_ (MNS) powder was obtained.

#### 2.2.2. Molding Process

HPC was prepared by wet mixing technology. Cement, FA, MNS, and sands were put into a compulsory mortar mixer and were dry mixed for 3 min. After that, uniform water with superplasticizer mixed was slowed added, and the fresh paste was mixed for another 3 min until the mixture enters the viscous flow state. After that, fresh pastes of HPC were cast in steel molds and were compacted to improve the density through a standard vibrating table. The specimens were demolded after 24 h of exposure under ambient condition, and were cured in the standard curing room (20 ± 2 °C, RH > 95%). Eventually, the mechanical properties and micro-properties of the samples were tested at different curing ages.

#### 2.2.3. Static Mechanical Strength

Specimens for static mechanical tests were 40 × 40 × 160 mm prisms, and flexural strength and compressive strength were tested according to Chinese test standard GB/T 17671-1999. These test procedures are the same to ASTM C348-14 and C349-14.

To guarantee the reliability of the test results, at each curing age, three samples with identical mix proportion were tested. The average value was served as the final flexural strength and compressive strength.

#### 2.2.4. Hydration Thermal Analysis

Heat of hydration was measured under isothermal conditions (20°C ± 0.1 °C) using an isothermal calorimeter produced by TAM instrument (TAM: New Castle, DE, USA; Model: TAM AIR). Each test consisted of a 15 g sample which was placed into the calorimeter cup. The cup was then put into the calorimeter and held for 5–6 h to attain temperature equilibrium. A data acquisition system was initiated at the same time to record the output voltage from which the heat flow in the system could be calculated. All tests were carried out for 72 h to observe later reactions.

#### 2.2.5. FTIR Test

Fourier transform infrared absorption spectroscopy (FTIR) (Bruker: Karlsruhe, Germany, Model: Nicolet5700) stimulates the energy level transition of molecules in the sample by irradiating the sample with infrared light. FTIR was used to characterize the functional groups on the surface of the nano-SiO_2_ particles. Firstly, powder samples were vacuum-dried for 48 h in a vacuum drying chamber at 50 °C. Secondly, sample powder and KBr powder were ground and mixed uniformly at the mass ratio of 1:150. Finally, the mixed powders were pressed into thin-sheet samples for testing. 

#### 2.2.6. DTA/TGA Test

For differential thermal analysis (DTA) and thermal gravimetric analysis (TGA) (NETZSCH: Selb, Germany, Model:STA449 F3), the specimens were soaked in absolute alcohol for 48 h to stop hydration and then milled to particles that could pass the 80 µm sieve. After that, the powders were dried in oven at 50 °C for 48 h. For the experiment, the heating rate was 10 °C /min and the temperature range was 25–900 °C.

#### 2.2.7. X-ray Diffraction Quantitative Analysis

For X-ray diffraction quantitative analysis (Bruker: Karlsruhe, Germany, Model:D8-Discover), the materials were soaked in absolute alcohol for 48 h to stop hydration and then milled to particles that could pass the 80 µm sieve. After that, the powders were dried in oven at 50 °C for 48 h. Finally, analytically pure α-Al2O3 and pre-specimen powers were mixed uniformly under the mass proportion of 1:9 for the quantitative analysis. The target-anode of X-ray diffraction was copper, and the working voltage and electric current were 40 kV and 30 mA, respectively. The step size was 0.02° 2θ. Scanning pace and scanning angular range were 0.30 s/step and 5–80°, respectively. Quantitative analysis was carried out by the software of Topas (Bruker AXS, Germany) which was based on the Rietveld method.

#### 2.2.8. SEM Test

For SEM analysis (FEI: Hillsboro, OR, USA, Model: FEI 3D), the specimen was cut into small sample of the size about 15 × 15 × 15 mm. Then, the small samples were embedded in epoxy resin for 24 h to complete hardening. After that, the sample surface was polished to make the surface smooth without scratches. Then, the samples were cleaned by ultrasonic in alcohol and dried in oven at 50 °C for 48. Finally, the sample was coated with gold to make it conductive before testing.

## 3. Results and Discussion

### 3.1. Modification Effect of Nano-SiO_2_ Particles

Figure 1 shows the FTIR spectra of NS and MNS. It can be found that there are absorption peaks at 3420 cm^−1^, 1636 cm^−1^ and 800 cm^−1^ in the spectra of both NS and MNS. The peak value at 800 cm^−1^ corresponds to the symmetrical stretching vibration of Si–O–Si, and the bending vibration occurs near 476 cm^−1^. The absorption peak at 1636 cm^−1^ corresponds to the bending of O–H and the absorption peak at 3420 cm^−1^ corresponds to the stretching of O–H.

Compared with the FTIR spectra of NS, the FTIR spectra of MNS have two more absorption peaks at 2850 cm^−1^ and 2930 cm^−1^, which correspond to the anti-symmetrical and symmetrical stretching vibration of –CH_2_–, respectively. There are two methylenes in the silane coupling agent of KH550, which also indicates that the silane coupling agent has been successfully grafted on the surface of nano-SiO_2_ particles.

The thermal gravimetric analysis results of nine groups of MNS are illustrated Figure 2. It can be seen from Figure 2 that there is a significant mass loss between 20 °C and 170 °C, which may be caused by the release of free water. There is no obvious mass loss between 170 °C and 300 °C due to the smooth curve. The mass loss also exists between 300 °C and 800 °C, which may be due to the separation of coupling agent grafted on the surface of nano-SiO_2_ particles. The curve tends to be flat after 800 °C, indicating that the decomposition is completed. In Figure 2, the decrease of each curve in the initial test stage is different, indicating that the water content in the sample is different. However, this paper focuses on the amount of coupling agents grafted on the surface of nano-SiO_2_ particles, so the mass loss rate of MNS between 300 and 800 °C can be used to roughly represent the grafting rate of the coupling agent.

Table 3 shows the mass loss rates of nine groups of MNS between 300 °C and 800 °C. It can be seen that the grafting rates of MNS2 and MNS3 are 4.32% and 4.47% respectively, indicating that the grafting effect is the best. Analysis result of the orthogonal Table 3 shows that the reaction temperature has the greatest influence on the grafting ratio of coupling agent, reaction time takes the second place, and the content of coupling agent has the least influence. Moreover, the grafting rate decreases with the increase of the coupling agent content, which may be due to excessive coupling agent and partial condensation of excessive KH550 hydrolysate, resulting in slow main reaction, thus reducing the grafting rate of modified nano-SiO_2_. When the reaction temperature is 65 °C, the grafting rate of the coupling agent is the highest. Too low temperature will slow down the reaction and too high will make the molecular movement too intense, the hydroxyl groups on the surface of nano-SiO_2_ may self-condensate and reduce the grafting sites, leading to the decrease of the grafting rate. Therefore, MNS2 is chosen as the best process in this paper. 

After cement hydration, the internal voids of cement paste are filled with alkaline solution with high PH value. Therefore, the saturated calcium hydroxide solution is used to simulate the internal void solution of cement to characterize the dispersion stability of NS and MNS in cement paste. The stability of NS and MNS in saturated calcium hydroxide are shown in Figure 3

It can be seen from Figure 3 that the obvious stratification of NS in saturated calcium hydroxide solution is due to the precipitation of large amount of nano-SiO_2_ particles, while MNS2 and MNS3 present a more turbid state in saturated calcium hydroxide solution, indicating that the modified nano-SiO_2_ particles have no obvious agglomeration or precipitation. The dispersibility of modified nano-SiO_2_ particles is obviously better than those before modification. This may be due to the higher steric hindrance of the modified nano-SiO_2_ particles. When the steric hindrance repulsion force exceeds van der Waals force, the modified nano-SiO_2_ particles are more difficult to agglomerate, which makes the modified nano-SiO_2_ particles show better dispersion in saturated calcium hydroxide solution than the unmodified nano-SiO_2_ particles. Gu [19] synthesized a series of nanoSiO_2_-polycarboxylate superplasticizer core-shell nanoparticles (nanoSiO_2_@PCE) from silanized polycarboxylate superplasticizer and colloidal nanoSiO_2_ by the ‘‘grafting to” method. The agglomeration of NS nanoparticles was reduced, that could be explained by a steric hindrance repulsive force created by the PCE shell, which would overcome the van der Waals attractive force between nanoparticles and promote the stability of NS@PCE. This is consistent with the results of this study.

### 3.2. Effects of MNS on Mechanical Properties of HPC

In order to reveal the influence of MNS on the mechanical properties of cement-based materials, the content of MNS was selected as 0.3%, 0.5%, and 0.7% respectively, and compared with the specimen incorporated with unmodified nano-SiO_2_ particles. The recipe of the HPC matrices is listed in Table 4 and the sand-binder ratios of all the five specimens were the same of 1.0. 

The compressive and flexural strengths at the ages of 3, 7, and 28 days were shown in Figure 4a,b, respectively. It can be observed that the compressive strength and flexural strength of different series of HPC materials show the same trend, that is, their strength increases with curing ages. As shown in Figure 4a, under the same curing age, compared with the reference specimen without nano-SiO_2_, the compressive strength of the material can be improved by adding NS and MNS, and the compressive strength of MNS is higher than that of NS. This may because of that MNS with better dispersibility can accelerate the hydration of cement and fill the internal voids of cement. Taking standard curing age of 7 days as an example, the compressive strength of the comparison specimen (NS-0%) is 62.2 MPa, and the strength of the specimens with 0.5% unmodified and modified nano-SiO_2_ is 65.3% and 69 MPa, which increase by 5% and 11% respectively. In addition, it can be seen that with the increase of MNS content, the compressive strength of the material has an upward trend, but the growth trend slows down. If the content is too high, the specific surface area of the system will increase accordingly, the water requirement will increase, and the hydration of cement particles will not be sufficient, which may have a negative impact on the mechanical properties of the composite. Figure 4b shows that the development of flexural strength is similar to that of compressive strength, but the increase of flexural strength by adding NS or MNS is limited.

### 3.3. Mechanism Analysis of HPC Reinforced by Modified Nano-SiO_2_

#### 3.3.1. Effects of MNS on Hydration

Figure 5 shows the effects of different dosages of NS and MNS on the rate of heat evolution of HPC at w/b of 0.2. The rate of heat evolution was calculated on the basis of a unit weight of plain Portland cement. It can be seen that the hydration heat curves of the five materials are relatively close. The hydration acceleration period of the specimens without nano-SiO_2_ is about 15–20 h, and the maximum peak value is reached at about 32 h. The addition of NS and MNS could both increase the hydration rate of cementitious composites and the time to reach the peak value is advanced by 2 h and 6 h, respectively. The cumulative heat released by adding MNS is also greater than that of NS in 3 days. It also proves that the MNS has better dispersibility which could promote the hydration of cement and accelerate the hydration process. This is consistent with the results of [19].

Collodetti [18] investigated the siloxane surface modified nano-SiO_2_ on heat development of Portland cement pastes, the results shown that the modified nanoparticles hindered the early hydration of cement. It was because of that strong retarder was used for functional groups on siloxane. Its purpose is to use the modified nano-materials as an additive to improve Portland cement paste hydration properties, but its modification method is worth learning.

#### 3.3.2. X-ray Diffraction Quantitative Analysis

In this research, the hydration products of the five kinds of specimens with different curing ages (1, 3, 7, and 28 d) are quantitatively analyzed. The quantitative analysis results of the main mineral phases of the specimens without nano-SiO_2_, with 0.5% NS and 0.5% MNS are listed in Table 5, respectively. Figure 6 gives an example of the quantitative result of MNS-0.5% for 7 days.

From the results of Table 5, it can be seen that all the three specimens have the similar trends: the unhydrated cement phase decreases and the amorphous phase of hydration products increases with the increase of curing age, and the hydration of Ca(OH)_2_ (CH) increases first and then continuously reduces. From the specimen without nano-SiO_2_, it can be seen that the cement is still at acceleration period at the curing age of 1 d, so the production of CH is less. At the end of 3 d, most of the cement hydration is completed, and the content of CH is the largest at this time. Then, due to the effect of fly ash, more amorphous phases are formed and CH is consumed. The content of CH is decreasing in the later period.

In addition, it can be seen that the amorphous phases increases and CH phase decreases with the adding of NS and MNS. This is due to that nan-SiO_2_ particles accelerated the hydration of cement and some CH reacted with SiO_2_ to form C–S–H phase. Besides, the content of CH in MNS specimen is lower than that in NS specimen at the same age, which indicates that more CH is consumed in the specimen of adding MNS. Without the adding of nano-SiO_2_ particles, the growth of C–S–H gel can only be restricted to the surface of cement particles. When adding smaller MNS, the C–S–H gel can also be formed on the surface of MNS particles. Due to its good dispersibility, non-agglomeration and smaller particle size, the nucleation point of C–S–H gel would be increased. The high reaction activity of MNS promotes the hydration of cement and reacts with CH to form C-S-H gels, in addition, the nucleation and super filling effect of MNS also make the structure of HPC material tend to be dense, thus showing excellent mechanical properties of HPC. 

#### 3.3.3. SEM Test Results

Figure 7 shows the SEM images of NS and MSN2. It can be seen from Figure 7a that there is a serious agglomeration between unmodified nano-SiO_2_ particles. The nano-SiO_2_ particles agglomerate into spheres, and the average size of agglomerated particles is about 1.52 μm. From Figure 7b, it can be seen that there is no obvious spherical aggregates. The modified nano-SiO_2_ particles are more dispersed and the size of a small amount of aggregates is less than 1 μm, which indicates that the dispersion performance of nano-SiO_2_ has been improved by the modification of KH550. This is mainly because of that the MNS prevents the condensation reaction of silane alcohol groups between different nano-SiO_2_ particles and effectively avoids the occurrence of agglomeration.

Figure 8 is the SEM image of the cement paste at the curing age of 28 days. By comparing Figure 8a1,b1, it can be found that there are many holes (the size is about 50 μm) and micro-cracks on the surface of the sample with unmodified nano-SiO_2_. The surface of the specimen with MNS has fewer holes and more compact structure. As can be seen from Figure 8b1, there are still a few agglomerated nano-SiO_2_ particles in the composites, which indicates that this part of nano-SiO_2_ particles are not involved in the process of cement hydration. However, it is difficult to find MNS in Figure 8b2. It is precisely because of the high dispersibility of MNS that its reinforcing effect can be brought into full play, so as to improve the micro-structure and mechanical properties of the composites. 

## 4. Conclusions

In this research, the effect of modified nano-SiO_2_ on the hydration process, microstructure, and mechanical property of HPC were investigated. The following conclusions can be drawn:

(1) The silane coupling agent (KH550) was used to modify the nano-SiO_2_ particles. The modification prevented the condensation reaction of silane alcohol groups between nano-SiO_2_ particles and effectively avoided the agglomeration. The optimum modification parameters were as follows: the coupling agent content of 10%, reaction temperature of 65 °C, and reaction time of 8 h.

(2) The MNS has better dispersibility in saturated calcium hydroxide solution. It may be due to the larger steric hindrance of the coupling agent grafted on the surface of nano-SiO_2_ particles, which prevents the agglomeration of particles.

(3) Compared with the reference specimen without nano-SiO_2_, the compressive strength of the composite can be improved by the adding NS and MNS at the same curing age. The compressive strength of specimen with MNS is better than that of NS. MNS with better dispersibility can accelerate cement hydration and fill in the internal voids of cement, thus making the microstructure of composites more compact and improving the compressive strength.

## Figures and Tables

**Figure 1 materials-13-00646-f001:**
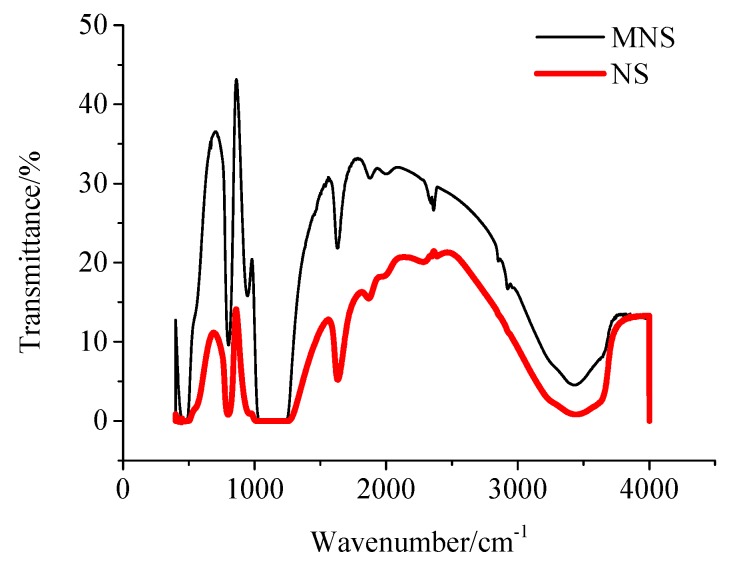
FITR curves for MNS and NS.

**Figure 2 materials-13-00646-f002:**
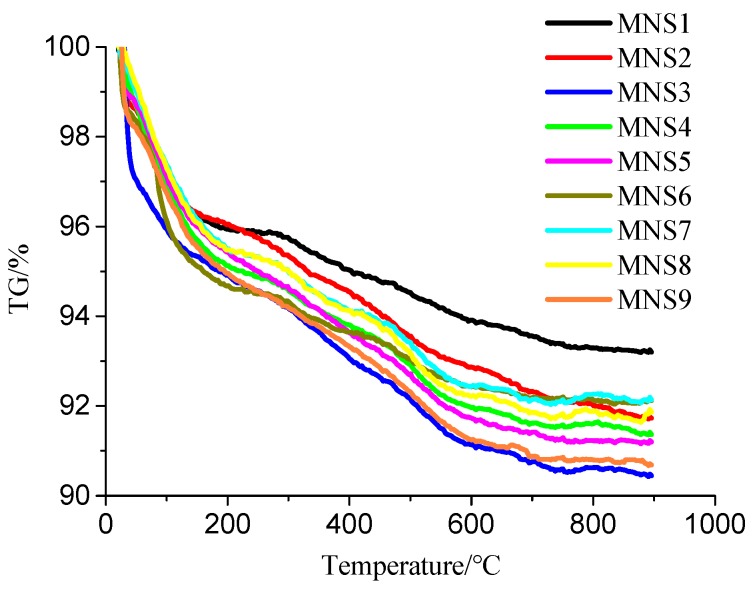
TG results of nine groups of MNS.

**Figure 3 materials-13-00646-f003:**
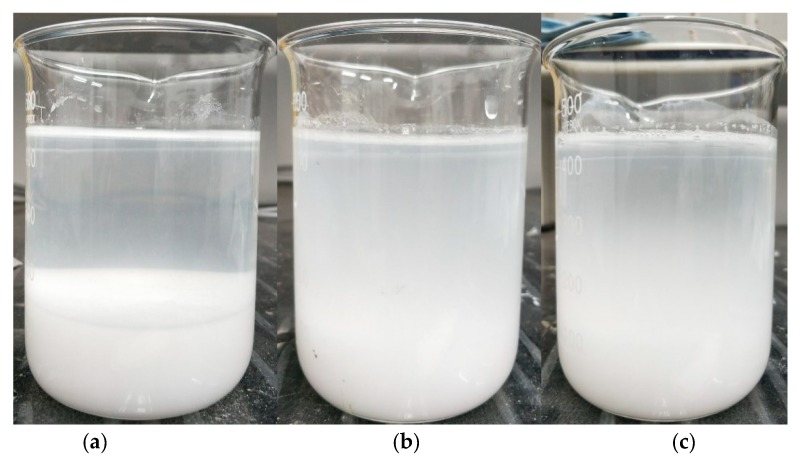
Stability of NS and MNS in saturated calcium hydroxide. (**a**) NS; (**b**) MNS2; (**c**) MNS3.

**Figure 4 materials-13-00646-f004:**
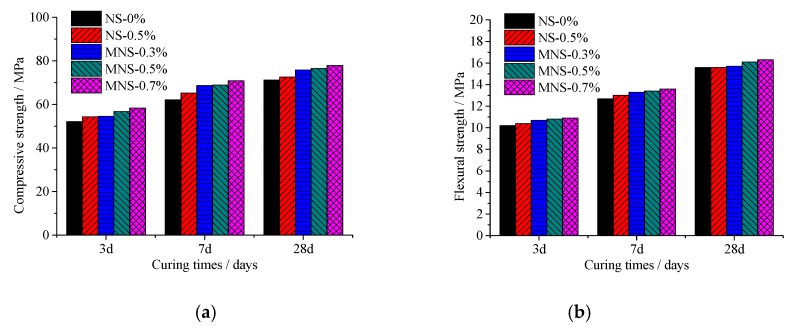
Compressive and flexural strengths of HPC at different curing ages. (**a**) Compressive strengths of HPC; (**b**) Flexural strengths of HPC.

**Figure 5 materials-13-00646-f005:**
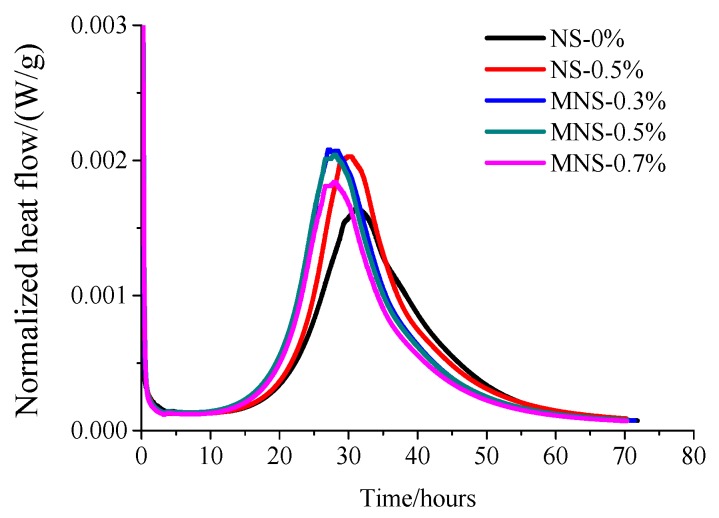
Heat evolution of HPCs with the addition of NS and MNS.

**Figure 6 materials-13-00646-f006:**
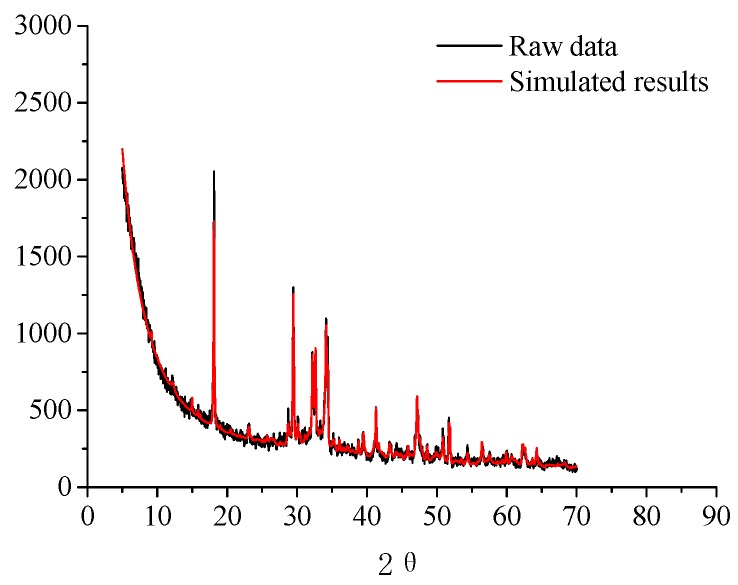
X-ray diffraction quantitative analysis result of MNS-0.5% (7 days).

**Figure 7 materials-13-00646-f007:**
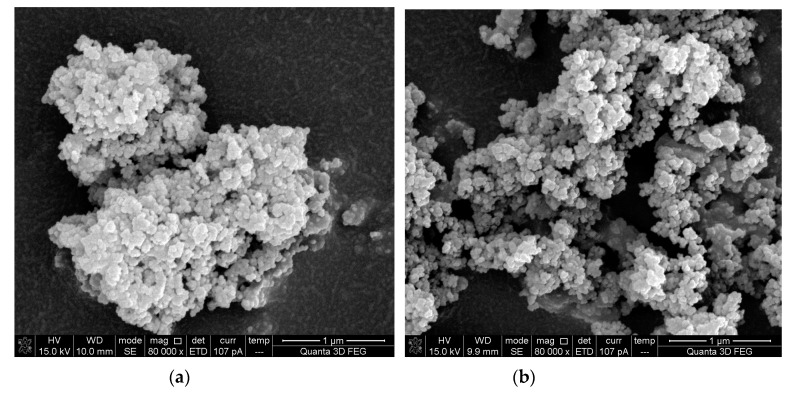
SEM micrographs of NS and MNS. (**a**) NS; (**b**) MNS.

**Figure 8 materials-13-00646-f008:**
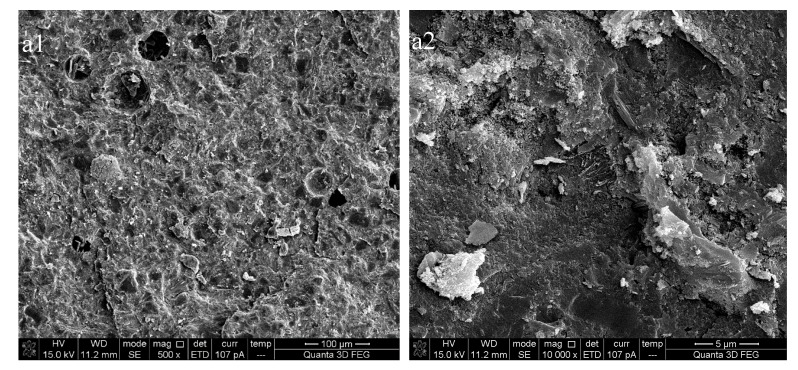
SEM micrographs of cement pastes. (**a1**) NS-0.5% (500×); (**a2**) NS-0.5% (10,000×); (**b1**) MNS-0.5% (500×); (**b2**) MNS-0.5%(10,000×).

**Table 1 materials-13-00646-t001:** Chemical composition of cement and fly ash (mass %).

Type	SiO_2_	Al_2_O_3_	Fe_2_O_3_	CaO	MgO	SO_3_	K_2_O	N_2_O	L.O.I
Cement	20.4	4.70	3.38	64.7	0.87	1.89	0.49	0.33	3.24
FA	53.98	28.84	6.49	4.77	1.31	1.16	1.61	1.03	0.72

**Table 2 materials-13-00646-t002:** Technological parameters of nano-SiO_2_ modification.

Items	Content of KH550 (%)	Temperature (°C)	Time (h)
MNS1	10	50	6
MNS 2	10	65	8
MNS 3	10	80	10
MNS 4	20	50	8
MNS 5	20	65	10
MNS 6	20	80	6
MNS 7	30	50	10
MNS 8	30	65	6
MNS 9	30	80	8

**Table 3 materials-13-00646-t003:** Grafting rate of nine orthogonal experiments.

Items	Content of KH550 (%)	Temperature (°C)	Times (h)	Grafting Rates (%)
MNS1	10	50	6	2.74
MNS 2	10	65	8	4.32
MNS 3	10	80	10	4.47
MNS 4	20	50	8	3.76
MNS 5	20	65	10	4.19
MNS 6	20	80	6	3.14
MNS 7	30	50	10	3.66
MNS 8	30	65	6	3.61
MNS 9	30	80	8	3.62

**Table 4 materials-13-00646-t004:** Mix proportions of HPC.

Items	Cement (%)	FA (%)	NS (%)	MNS (%)	w/b	SuperplasTicizer (%)
NS-0%	65	35	0	0	0.2	2
NS-0.5%	64.5	35	0.5	0	0.2	2
MNS-0.3%	64.7	35	0	0.3	0.2	2
MNS-0.5%	64.5	35	0	0.5	0.2	2
MNS-0.7%	64.3	35	0	0.7	0.2	2

**Table 5 materials-13-00646-t005:** Quantitative analysis results of HPC at different curing times (%).

Ages (d)	Item	Mineral Composition					
		C_3_S	C_2_S	C_3_A	C_4_AF	Ca(OH)_2_	Amorphous Phase
1	NS-0%	14.7	8.5	2.5	3.8	2.2	46.5
	NS-0.5%	13.8	8.1	2.2	3.4	1.9	48.3
	MNS-0.5%	11.6	7.2	1.8	2.8	2.1	50.2
3	NS-0%	10.6	6.9	2.1	3.3	4.8	53.6
	NS-0.5%	9.9	5.7	1.8	2.6	3.5	54.8
	MNS-0.5%	8.6	5.3	1.2	2.3	3.1	55.7
7	NS-0%	5.6	5.6	1.3	3.1	3.8	57.3
	NS-0.5%	5.1	4.5	0.9	2.4	3.2	58.4
	MNS-0.5%	4.5	4.1	0.7	2.2	2.9	59.3
28	NS-0%	5.0	5.1	1.1	2.8	3.4	59.1
	NS-0.5%	4.7	4.3	0.9	2.1	2.9	59.6
	MNS-0.5%	4.3	4.1	0.6	1.8	2.5	59.7

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
