# Peer review of "Effects of Modified Nano-SiO2 Particles on Properties of High-Performance Cement-Based Composites"

_materials, 2020, doi:10.3390/ma13030646_

Round 1

Reviewer 1 Report

The manuscript “Effects of modified nano-SiO2 particles on properties of high performance cement-based composites” brings some interesting developments on the application of SiO2 particles in concrete.

Authors should clarify in the introduction any limitation of using modified nano-SiO2 particles in large scale projects. Would it be feasible to produce such particles in large scale?

Clarify the number of samples used for compressive and flexural strength tests  

Clarify how the quantitative XRD analysis were done –  software, technique used.

Author Response

This paper adopts a simple and effective surface modification method, which is more suitable for large scale production.

To guarantee the reliability of the test results, at each curing age, three samples with identical mix proportion were tested.

Quantitative analysis were carried out by the software of Topas which was based on the Rietveld method.

Reviewer 2 Report

The originality of the paper is relatively low and insufficiently presented in the context of the recent studies. The state-of-art is unproperly presented, for instance the article https://doi.org/10.1016/j.conbuildmat.2017.12.159 is ignored while the findings of the second one: https://doi.org/10.1016/j.conbuildmat.2018.10.214 are marginalised.

In general, the scientific results and conclusions are insufficiently described particularly in the light of the recent findings of other authors.

Moreover, statistical analysis of the experimental results is not provided.

Additional comments:

6 line 102 - remove brand names of the apparatus 7 Fig.1 – MNS curve is not visible; the figure needs correction 8 Fig.2 – the behaviour of MNS 3 and MNS 1 is different while almost no comment is provided by the authors in this context.

Author Response

Comparative analysis with the recent findings of other authors has been added to the research results.

In this paper, more than three samples are used in the tests, including static mechanical properties, hydration heat, FTIR test, XRD, etc. The results are statistically analyzed and repeatable.

The brand names of the apparatus have been removed according to the suggesting.

Fig.1 has been corrected according to the suggesting.

In Fig.2, the behavior of MNS 3 and MNS 1 is different, especially in the initial test stage, indicating that the water content in the sample is different. However, this paper focuses on the amount of coupling agents grafted on the surface of nano-SiO2 particles.

Reviewer 3 Report

The authors have investigated the effect of modifying surface of silicon dioxide nanoparticles on performance of cement-based composites, particularly their static mechanical properties. FTIR results show that the silane coupling agent that was used as surface modifier of silicon dioxide nanoparticle, has successfully grafted on its surface. Compressive strength and flexural strength of the cement based composites were tested for curing times of 3, 7 and 28 days. These properties were improved for samples with modified nanoparticles for the investigated curing periods. According to SEM results, dispersity of the modified nanopartcles were improved, this was evident from the particle size for both modified and unmodified samples. Overall, the findings in this study have merit and I recommend acceptance of the article upon making one revision, which is to explain the procedure for evaluating the compressive and flexural strength.

Author Response

The language and structure of whole manuscript have been carefully checked.

Reviewer 4 Report

A silane coupling agent is used to adjust the mechanical properties of high-performance cement-based composites.  The results presented by the authors supported their claims.  There are several minor styles and punctuation issues in the manuscript that need to be fixed before accepting the paper. 

The apparent results are not needed to be written in the abstract (e.g., Line 8 and 9: “The results indicated that the dispersibility of modified nano-SiO2 particles was obviously improved.” The style of paragraph headings should be changed in the method section. The current version is very confusing given that the title of the section and the section text is only separated by a few white spaces.  The authors must pay attention to the punctuation marks. Many places punctuation marks are missing (e.g., Line 291).

Author Response

The whole manuscript have been carefully checked.

The abstract has been revised according to the suggesting.

The style of paragraph headings has been changed in the method section.

The punctuation marks in this paper have been carefully checked and revised.
